# Analytical Performances of the COVISTIX^TM^ Antigen Rapid Test for SARS-CoV-2 Detection in an Unselected Population (All-Comers)

**DOI:** 10.3390/pathogens11060628

**Published:** 2022-05-30

**Authors:** Francisco Garcia-Cardenas, Fernando Peñaloza, Jennifer Bertin-Montoya, Rafael Valdéz-Vázquez, Alba Franco, Ricardo Cortés, Emmanuel Frias-Jimenez, Alberto Cedro-Tanda, Alfredo Mendoza-Vargas, Juan Pablo Reyes-Grajeda, Alfredo Hidalgo-Miranda, Luis A. Herrera

**Affiliations:** 1Instituto Nacional de Medicina Genómica, Periférico Sur 4809, Arenal Tepepan, Tlalpan, Mexico City 14610, Mexico; franciscojavgc@gmail.com (F.G.-C.); fpenaloz@icloud.com (F.P.); alba.franco.13@gmail.com (A.F.); richgomezcortes@gmail.com (R.C.); jfrias@inmegen.gob.mx (E.F.-J.); acedro@inmegen.gob.mx (A.C.-T.); amendoza@inmegen.gob.mx (A.M.-V.); jreyes@inmegen.gob.mx (J.P.R.-G.); 2Centro Citibanamex COVID, Mexico City 11200, Mexico; jennybertin@yahoo.com (J.B.-M.); rrvaldezvazquez@gmail.com (R.V.-V.); 3Unidad de Investigación Biomédica en Cáncer, Instituto Nacional de Cancerología, Instituto de Investigaciones Biomédicas, Universidad Nacional Autónoma de México, Mexico City 04510, Mexico

**Keywords:** COVID-19 detection, surveillance, antigen rapid test

## Abstract

The performance and validity of the COVISTIX^TM^ rapid antigen test for the detection of SARS-CoV-2 were evaluated in an unselected population. Additionally, we assessed the influence of the Omicron SARS-CoV-2 variant in the performance of this antigen rapid test. Swab samples were collected at two point-of-care facilities in Mexico City from individuals that were probable COVID-19 cases, as they were either symptomatic or asymptomatic persons at risk of infection due to close contact with SARS-CoV-2 positive cases. Detection of the Omicron SARS-CoV-2 variant was performed in 91 positive cases by Illumina sequencing. Specificity and sensitivity of the COVISTIX^TM^ rapid antigen test was 96% (CI 95% 94–98) and 81% (CI 95% 76–85), respectively. The accuracy parameters were not affected in samples collected after 7 days of symptom onset, and it was possible to detect almost 65% of samples with a Ct-value between 30 and 34. The COVISTIX^TM^ antigen rapid test is highly sensitive (93%; CI 95% 88–98) and specific (98%; CI 95% 97–99) for detecting Omicron SARS-CoV-2 variant carriers. The COVISTIX^TM^ rapid antigen test is adequate for examining asymptomatic and symptomatic individuals, including those who have passed the peak of viral shedding, as well as carriers of the highly prevalent Omicron SARS-CoV-2 variant.

## 1. Introduction

The coronavirus disease 2019 (COVID-19) pandemic has exerted unprecedented effects on healthcare and economic systems globally. A steady increase in severe acute respiratory syndrome coronavirus 2 (SARS-CoV-2) cases worldwide is causing some regions of the world to withstand a third or even fourth wave of contagion [1,2]. Swift detection of SARS-CoV-2 infection is paramount for the containment of cases, for the prevention of sustained contagion, for the return to economic and education activities, and most importantly, for the reduction of mortality [3,4].

Reverse transcription polymerase chain reaction (RT-PCR) is considered the gold standard method for detecting SARS-CoV-2 infection due to its high sensitivity and specificity [5,6]. However, implementing RT-PCR testing in massive screening campaigns requires specialized protective equipment, qualified personnel, and sample transportation to a centralized laboratory, which has proven to be challenging, particularly in resource-limited settings [2]. As a result of these limitations, several rapid tests based on SARS-CoV-2 antigen detection by immunochromatography have been introduced, offering improved access to testing due to faster result availability, simple use at point-of-care, and low costs. Rapid antigen tests represent an appealing alternative for large-scale testing of the general population [7].

As an essential part of the public health response to the COVID-19 pandemic, Mexico City’s government implemented a surveillance strategy intended to detect active cases among the general population, which was initially based on RT-PCR. In November 2020, rapid antigen tests were also included in the strategy. According to data reported by Mexico City’s Digital Agency of Public Innovation, almost 1 million antigen tests were performed by the beginning of February 2021 [8].

The World Health Organization (WHO) recognizes that although antigen tests have proven lower sensitivity than molecular tests, they provide rapid and less resource-consuming means of detection of SARS-CoV-2 in individuals who have high viral loads and therefore have a higher risk of disease transmission [9]. Currently, Mexico’s Institute of Epidemiologic Reference and Diagnosis (InDRE) has evaluated and approved more than 20 rapid antigen tests used for SARS-CoV-2 detection, including Sorrento’s COVISTIX^TM^ COVID-19 Ag Rapid Test Device [10]. These single-use devices are immunochromatographic assays that detect the SARS-CoV-2 nucleocapsid protein and provide results of active infection within 20 min; thus, they are faster than a RT-PCR procedure [7].

Different strategies for mitigating contagion using antigen rapid device tests have been proposed, such as the use only in symptomatic patients within 5–7 days after symptom onset when the viral load is at its peak. Furthermore, strategies such as the implementation of these assays to detect SARS-CoV-2 in healthcare workers and contacts of confirmed cases could be beneficial for pandemic containment [7]. Therefore, the need has emerged for an economic yet precise testing strategy and a more sensitive antigen test that can rapidly detect lower viral loads at the very beginning of the disease or after 8 days after the onset of symptoms.

Here, we set to evaluate the performance and validity of Sorrento’s COVISTIX^TM^ rapid antigen test for the detection of SARS-CoV-2 in a Mexican open population through nasal and nasopharyngeal swabs and compare it to RT-PCR. Additionally, we evaluate the performance of the COVISTIX^TM^ rapid antigen test to detect individuals infected with the Omicron variant.

## 2. Results

A total of 783 subjects were included to evaluate the COVISTIX^TM^ assay, 254 of which had a positive RT-PCR test (prevalence 32.4%). In this group of individuals, 391 were female and 392 were male. The median (IQR) age in years was 40 (28–51). Nasal and nasopharyngeal samples were evaluated with the COVISTIX^TM^ rapid antigen test and compared to a nasopharyngeal swab analyzed with RT-PCR. Table 1 shows that out of 783 samples, 205 tested positive both by RT-PCR and by COVISTIX^TM^ and 508 were detected as negative by both assays, showing false-negative results in 49 samples (19.3%) and 21 false-positive results (4%). Overall specificity and sensitivity of the COVISTIX^TM^ rapid antigen test were 96% (CI 95%: 94–98) and 81% (CI 95%: 76–85), respectively. Positive and negative likelihood ratios were 20.25 (CI 95%: 13.0–31.0) and 0.2 (CI 95%: 0.16–0.26) each. Positive post-test probability was 91% and negative post-test probability was 12%. Cohen’s kappa coefficient shows a very good concordance between results obtained by COVISTIX^TM^ and RT-PCR (0.8; CI 95%: 0.72–0.86). 

The performance of the COVISTIX^TM^ rapid antigen test based on the RT-PCR Ct-value shows that although the assay’s sensitivity was higher in samples with a Ct-value below 30, as it has been reported for other rapid antigen tests, the COVISTIX^TM^ assay detected almost 65% of SARS-CoV-2 carriers with Ct-values between 30 and 34 (Table 2).

We also analyzed the performance of the COVISTIX^TM^ assay in symptomatic (N = 335) and asymptomatic (N = 448) individuals. Results showed an overall sensitivity that was similar among symptomatic individuals regardless of the number of days after symptom onset (Table 3A,B). When we compared the sensitivity among symptomatic individuals according to the Ct-value, we did not observe significant differences in the sensitivity of the COVISTIX^TM^ assay in individuals with Ct-values below 30 (*t*-test for J indexes: 0.15). The sensitivity observed in the asymptomatic group was lower than in the symptomatic group (Table 3C); nevertheless, the difference was not significant when we compared the J indexes of both groups (*t*-test: 0.30).

A second cohort was recruited and tested between 15 December 2021 and 5 January 2022 during the fourth wave of the pandemic in Mexico. Out of 999 samples, 131 were positive by RT-PCR. The overall sensitivity and specificity of the COVISTIX^TM^ antigen rapid test were 72% and 98%, respectively, and the test accuracy was 0.95 (CI 95%: 0.93–0.96). The performance of the COVISTIX^TM^ antigen rapid test was higher if only RT-PCR positive samples with Ct-values of 34 or less were included in the evaluation (N = 103), showing a sensitivity and specificity of 92% and 98%, respectively, and a test accuracy of 98% (CI 95%: 96–98%). 

Since the SARS-CoV-2 variant, Omicron, was highly prevalent worldwide during that period, we investigated whether the presence of this variant has an impact on the diagnostic performance of the COVISTIX^TM^ assay. Out of 951 individuals, 91 were identified as SARS-CoV-2 positive with a viral Ct-value ≤ 30 between 15 December 2021 and 5 January 2022. All of them were confirmed as Omicron BA.1 carriers by sequencing. The diagnostic performance of the COVISTIX^TM^ antigen rapid test showed that the overall sensitivity and specificity of the COVISTIX^TM^ antigen rapid test were high, being 93% and 98%, respectively, as was the test accuracy: 98% (CI 95%: 97–99%) (Table 4). We did not detect significant differences in the sensitivity of this antigen rapid test associated with Ct-values below 30.

## 3. Discussion

The results show a very good performance of the COVISTIX^TM^ antigen rapid test to detect SARS-CoV-2 infections in an all-comers general population. To evaluate the overall accuracy of the COVISTIX^TM^ assay for detection of SARS-CoV-2 carriers, we calculated the Youden index (J), which is a useful measure of the misclassification error in diagnostic tests [11]. The J value for COVISTIX^TM^ was 0.77 (CI 95%: 0.72–0.82; SE: 0.026), indicating a good performance of this assay. 

Several reports have emphasized the importance of the viral load for the detection of SARS-CoV-2 carriers by rapid antigen tests, which are best-suited for the rapid identification of individuals carrying high viral loads [12,13]. The RT-PCR Ct-value is considered a surrogate parameter for viral load; the lower the Ct-value, the higher the expected viral load. The results indicate that the COVISTIX^TM^ assay is effective to detect not only highly infectious individuals, but also those potentially carrying low viral loads. 

A major concern around the use of rapid antigen tests for massive screening or even for diagnosis is the elevated frequency of false negatives, whose impact in pandemic control could be detrimental since false-negative individuals could spread the virus due to an unjustified sense of security [14]. In general, several studies suggest that rapid antigen tests are frequently negative in RT-PCR positive samples with Ct-values above 29, which could lead to an elevated number of undetected SARS-CoV-2 carriers; since there is no minimal infectious dose reported to date, it cannot be assumed that individuals whose samples report Ct-values above 30 are not contagious [15,16]. In fact, La Scola et al. reported that 50% of clinical specimens with a Ct-value equal to or more than 30 can be cultured and be potentially infectious [17]. 

The WHO recommends screening using rapid antigen tests only in cases where the pre-test probability is greater than 5% (9); in this study, the pre-test probability in asymptomatic individuals was 13.6%, and the results show that the performance of the COVISTIX^TM^ rapid antigen test is adequate even if sampling is not restricted to individuals within the first seven-day period of symptom onset nor to individuals with a low viral load. False-positive samples that were RT-PCR negative but positive from the COVISTIX^TM^ test were detected in both cohorts; all these individuals were considered as potentially positive cases until confirmed with a second RT-PCR test that was performed after 3 days. Results show that all samples were confirmed as RT-PCR negative, thus the individuals were considered as SARS-CoV-2 negative. A potential explanation for these discordant results is the presence of the virus’s proteins or peptides in the respiratory tract, which could be reactive for the antigen test. 

Emergence and spread of new SARS-CoV-2 variants might have an impact on the performance of diagnostic tests, such as rapid antigen tests. Given the high prevalence of the Omicron variant in Mexico City, we also evaluated the performance of the COVISTIX^TM^ assay on an additional cohort which included 999 samples collected between 15 December 2021 and 5 January 2022. By the end of the collection period, prevalence of Omicron was 87.7% in the samples sequenced at INMEGEN. The overall sensitivity and specificity of the antigen rapid test in the 999 samples cohort were 72% and 98%, respectively, and the test accuracy was 0.95 (CI 95%: 0.93–0.96). The performance of the test was corroborated in this cohort with samples that had Ct-values of 34 or less (N = 103), showing a sensitivity and specificity of 92% and 98%, respectively, and a test accuracy of 98% (CI 95%: 96–98%). When we focused on samples where Omicron was confirmed by sequencing (N = 91), all samples were determined as carriers of the lineage BA.1; the overall sensitivity and specificity of the test were 93% and 98%, respectively, with a test accuracy of 98% (CI 95%: 97–99%, Table 4). This data indicates that the performance of the COVISTIX^TM^ assay is not affected by the presence of the Omicron variant.

Our results indicate that the COVISTIX^TM^ rapid antigen test is highly sensitive and specific for identifying SARS-CoV-2 carriers, even current variants such as Omicron; thus, it is suitable for testing populations that return to presential activities as this test can reduce the contagion risk by efficiently detecting infected individuals even with low viral loads.

## 4. Materials and Methods

### 4.1. Respiratory Specimens

This study was conducted by the National Institute of Genomic Medicine of Mexico (INMEGEN), in collaboration with the Citibanamex COVID temporal unit set in Mexico City. Samples from 783 individuals were collected from the Citibanamex COVID temporal unit and from INMEGEN, Mexico City between 1 May and 16 August 2021. Samples from a second cohort (N = 999) were collected between 15 December 2021 and 5 January 2022 to analyze the performance of the COVISTIX^TM^ rapid antigen test at detecting individuals infected with the SARS-CoV-2 Omicron variant. SARS-CoV-2 PCR-positive samples were selected for inclusion in this analysis based on viral Ct ≤ 30 (N = 91).

Participants defined as all-comers, in both cohorts, were any person that requested a test for SARS-CoV-2 at one of the two points-of-care mentioned above. Individuals were sampled and tested independently of their symptoms.

Nasal and nasopharyngeal swabs were obtained from each individual and tested with the COVISTIX^TM^ rapid antigen test, as well as with RT-PCR. All individuals were evaluated with the COVISTIX^TM^ rapid antigen test following the algorithm: An initial nasal swab was tested with COVISTIX^TM^; if negative, a nasopharyngeal swab was taken and tested with COVISTIX^TM^. The dedicated nasopharyngeal swab was inserted by trained clinicians through the nostril parallel to the palate to a depth equal to the distance from nostril to outer opening of ear. The swab was rolled four times and left in place for several seconds. The swab was then removed and inserted into the extraction tube. Individuals were considered as positive for SARS-CoV-2 if either the nasal or the nasopharyngeal swab resulted positive with the COVISTIX^TM^ rapid antigen test. 

Results were confirmed by RT-PCR in a nasopharyngeal sample taken in parallel. Nasopharyngeal swabs were collected by a trained clinician with a flexible nylon swab that was inserted through the patient’s nostrils to reach the posterior nasopharynx. It was left in place for several seconds and slowly removed while rotating. The swab was then placed in 3 mL of sterile viral transport media. Swabs from two nostrils were deposited in a single viral transport tube and kept at 4 °C until their transportation, which was within the next 4 h, to the processing laboratory at INMEGEN for viral RNA extraction and SARS-CoV-2 detection.

Individuals with a RT-PCR negative result that were positive with the COVISTIX^TM^ test were considered as potentially positive cases, until confirmation with a second RT-PCR test was performed after 3 days (Table 5).

### 4.2. SARS-CoV-2 Rapid Antigen Test

Antigen tests were performed on-site using the COVISTIX^TM^ rapid antigen test (Sorrento Therapeutics, San Diego, CA, USA), which is a lateral flow immunoassay for rapid SARS-CoV-2 antigen detection. COVISTIX^TM^ detects single SARS-CoV-2 virus antigens directly from either a shallow nasal or nasopharyngeal swab sample in 15–20 min. COVISTIX™ uses a simple one-step protocol that only requires mixing the sample with a buffer and applying the sample to the COVISTIX™ test well. The test uses a proprietary platinum nano-catalyst core (PtNC) which yields up to 100-fold increases in sensitivity over conventional lateral flow colloidal gold assays. As viral antigen passes over the labeled antibody, the sample encounters the PtNC particles targeted specifically to the virus nucleocapsid (N) or matrix (M) antigens. The antigen is then captured in a strong biotin-avidin complex, producing a conspicuous black line on the membrane stick and indicating a positive result. 

### 4.3. RNA Extraction and RT-PCR SARS-CoV-2 Detection

Total nucleic acids were extracted from 200 μL of viral transport media from the swab using the MagMAX Viral/Pathogen Nucleic Acid Isolation Kit (Thermo Fisher Scientific, Life Technologies Corporation, Austin, TX, USA), following the manufacturer’s instructions, and eluted into 75 μL of elution buffer. RT-PCR was performed using the TaqPath™ COVID-19 CE-IVD RT-PCR Kit (Thermo Fisher Scientific, Life Technologies Corporation, Pleasanton, CA, USA), following manufacturer’s instructions [18]. Briefly, the method can detect three specific genes of SARS-CoV-2, along with an internal positive control in a single PCR reaction that targets the exogenous control RNA of the MS2 bacteriophage. The kit detects the ORF1ab, S, and N genes of the virus. We classified samples as positive for SARS-CoV-2 when at least two genes were detected with a Ct-value of 37 or less. If only one of these genes was detected, we labeled the sample as inconclusive. We ran all tests with Thermo Fisher’s ABI QuantStudio 5 or QuantStudio 7 real-time thermal cyclers.

### 4.4. Illumina Sequencing

SARS-CoV-2 PCR-positive samples were selected for inclusion in this analysis based on a viral Ct ≤ 30 (N = 91). The libraries were prepared using the Illumina COVID-seq protocol, following the manufacturer’s instructions. First-strand synthesis was carried out on RNA samples. The synthesized cDNA was amplified using ARTIC primers V3 for multiplex PCR, generating 98 amplicons across the SARS-CoV-2 genome [19]. The PCR-amplified product was tagmented and adapted using IDT for Illumina Nextera UD Indices Set A, B, C, D (384 indices) (Illumina, San Diego, CA, USA). Dual-indexed pair-end sequencing with a 36 bp read length was carried out on the NextSeq 550 platform (Illumina, San Diego, CA, USA).Illumina raw data were processed using DRAGEN Lineage v3.3.4/.5 with standard parameters (Illumina, San Diego, CA, USA). Further samples with SARS-CoV-2 and at least 90 targets detected were processed for lineage designation.

### 4.5. Statistical Analysis

A 2 × 2 table was built using RT-PCR as the gold standard. Sensitivity, specificity, likelihood ratios, post-test probabilities (http://araw.mede.uic.edu/cgi-bin/testcalc.pl accessed on 27 April 2022. Cohen’s *kappa* correlation coefficient, the Youden index and accuracy were calculated for the COVISTIX^TM^ rapid antigen test.

### 4.6. Patient and Public Involvement

Patients or the public were not involved in the design, conduct, reporting, or dissemination plans of our research.

## Figures and Tables

**Table 1 pathogens-11-00628-t001:** Results of COVISTIX^TM^ SARS-CoV-2 antigen rapid test in swabs from 783 individuals (all-comers) compared to RT-PCR.

Sex	Female: 391 (49.9%)		Male: 392 (50.1%)
Median Age	40 Years (IQR: 28–51)		
	PCR (+)	PCR (−)	Total
COVISTIX (+)	205	21	226
COVISTIX (−)	49	508	557
Total	254	529	783
Prevalence	0.32		
Sensitivity	0.81 (CI 95%: 0.75–0.85)		
Specificity	0.96 (CI 95%: 0.94–0.98)		
LR (+)	20.25 (CI 95%: 13–31)		
LR (−)	0.2 (CI 95%: 0.16–0.26)		
Post-test (+)	0.91 (CI 95%: 0.86–0.94)		
Post-test (−)	0.12 (CI 95%: 0.07–0.11)		
Kappa	0.8 (CI 95%: 0.72–0.86)		
Youden index	0.77 (CI 95%: 0.72–0.82; SE: 0.026)		
Accuracy	0.91 (CI 95%: 0.89–0.93)		
LR: likelihood ratio; Post-test: post-test probability. Kappa: Cohen’s kappa correlation (very good)			

**Table 2 pathogens-11-00628-t002:** COVISTIX^TM^ results compared to RT-PCR (N = 783) depending on the Ct-value.

RT-PCR (+)	COVISTIX (+)	COVISTIX (−)	Sensitivity (CI 95%)		
254	205	49	0.81 (0.75–0.85)		
Ct ≥ 34 = 25	8	17	0.32		
30 ≥ Ct < 34 = 33	21	12	0.64		
25 ≥ Ct < 30 = 66	53	13	0.80		0.90
20 ≥ Ct < 25 = 75	70	5	0.93	0.95	
Ct < 20 = 55	53	2	0.96		
RT-PCR (−)	COVISTIX (+)	COVISTIX (−)	Specificity (CI 95%)		
529	21	508	0.96 (0.94–0.98)		

**Table 3 pathogens-11-00628-t003:** Overall performance of the COVISTIX^TM^ rapid antigen test in symptomatic (N = 335) and asymptomatic (N = 448) individuals. (**A**) Individuals ≤ 7 days of symptom onset (N = 240); (**B**) individuals > 7 days of symptom onset (N = 95); (**C**) asymptomatic individuals (N = 448).

(**A**)					
RT-PCR (+)	COVISTIX (+)	COVISTIX (−)	Sensitivity (CI 95%)		
125	103	22	0.82 (0.75–0.89)		
Ct ≥ 34 = 10	3	7	0.30		
30 ≥ Ct < 34 = 7	4	3	0.57		
25 ≥ Ct < 30 = 30	23	7	0.77		0.89
20 ≥ Ct < 25 = 46	43	3	0.94	0.94	
Ct < 20 = 32	30	2	0.94		
RT-PCR (−)	COVISTIX (+)	COVISTIX (−)	Specificity (CI 95%)		
115	12	103	0.90 (0.82–0.94)		
(**B**)					
RT-PCR (+)	COVISTIX (+)	COVISTIX (−)	Sensitivity (CI 95%)		
68	56	12	0.82 (0.71–0.90)		
Ct ≥ 34 = 11	6	5	0.55		
30 ≥ Ct < 34 = 20	15	5	0.75		
25 ≥ Ct < 30 = 23	21	2	0.91		0.95
20 ≥ Ct < 25 = 7	7	0	1.00	1.00	
Ct < 20 = 7	7	0	1.00		
RT-PCR (−)	COVISTIX (+)	COVISTIX (−)	Specificity (CI 95%)		
27	4	23	0.85 (0.66–0.96)		
(**C**)					
RT-PCR (+)	COVISTIX (+)	COVISTIX (−)	Sensitivity (CI 95%)		
61	46	15	0.75 (0.63–0.85)		
Ct ≥ 34 = 5	0	5	0.0		
30 ≥ Ct < 34 = 5	1	4	0.20		
25 ≥ Ct < 30 = 13	9	4	0.69		0.88
20 ≥ Ct < 25 = 22	20	2	0.91	0.95	
Ct < 20 = 16	16	0	1.00		
RT-PCR (−)	COVISTIX (+)	COVISTIX (−)	Specificity (CI 95%)		
387	5	382	0.99 (0.97–1.00)		

**Table 4 pathogens-11-00628-t004:** Results of COVISTIX^TM^ SARS-CoV-2 antigen rapid test from swabs of 91 Omicron positive individuals (all-comers) as compared to RT-PCR. Out of 951 individuals, 91 tested positive between 15 December 2021 and 5 January 2022. SARS-CoV-2 PCR-positive samples were selected for inclusion in this analysis based on viral Ct ≤ 30 (N = 91).

	PCR (+)	PCR (−)	Total
COVISTIX (+)	85	15	100
COVISTIX (−)	6	845	851
Total	91	860	951
Prevalence	0.096		
Sensitivity	0.93 (CI 95%: 0.88–0.98)		
Specificity	0.98 (CI 95%: 0.97–0.99)		
LR (+)	0.54 (CI 95%: 0.32–0.89)		
LR (−)	0.07 (CI 95%: 0.03–0.15)		
Post-test (+)	0.85 (CI 95%: 0.77–0.90)		
Post-test (−)	0.01 (CI 95%: 0.00–0.02)		
Kappa	0.88 (CI 95%: 0.81–0.94)		
Youden index	0.92 (SE: 0.026)		
Accuracy	0.98 (CI 95%: 0.97–0.99)		
LR: likelihood ratio; Post-test: post-test probability. Kappa: Cohen’s kappa correlation (very good)			

**Table 5 pathogens-11-00628-t005:** Interpretation criteria for diagnosis of individuals and validation of samples.

COVISTIX^TM^		RT-PCR	Interpretation	Observations
Nasal	NPS			
+	NA	+	Positive	
-	+	+	Positive	
+	−	−	Potential Positive	A sample taken 3 days after must be considered for confirmatory RT-PCR
−	+	−	Potential Positive	A sample taken 3 days after must be considered for confirmatory RT-PCR
−	−	−	Negative	

## Data Availability

No additional data available.

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
