# Peer review of "Analytical Performances of the COVISTIXTM Antigen Rapid Test for SARS-CoV-2 Detection in an Unselected Population (All-Comers)"

_pathogens, 2022, doi:10.3390/pathogens11060628_

Round 1

Reviewer 1 Report

The manuscript proposed by Garcia-Cardenas at al. is intended to report the performance of the COVISTIX antigen rapid test to detect SARS-CoV-2 infections in all-comers general population. The authors reported a statistical analysis of rapid antigen test output comparing it to the gold standard RT-PCR. They focused the attention of different types of comers regarding the symptom’s onset or absence as well as Ct values of RT-PCR output. Illumina sequencing about omicron variant is an added value.

Minor revisions

The general formatting of the file should be improved.

Lines 36-37: “acute respiratory syndrome coronavirus 2 (SARS-CoV-2)” should be changed in “Severe Acute Respiratory Syndrome Coronavirus 2 (SARS-CoV-2)”.

Line 65: “SARS-COV-2” should be changed in “SARS-CoV-2”

Line 119: “forth” should be changed in “fourth”

Line 201: “The swab was the removed” should be changed in “The swab was then removed”

Major revisions

My major concerns are:

At line 118 the author mentioned that “One-hundred thirty-one out of 999 samples tested between December 15th 2021 and January 5th 2022, during the forth wave of pandemic in Mexico, were positive by RT-PCR”. This sentence is in contrast with the following sentence (line 122) “The performance of the COVISTIXTM antigen rapid test is higher if only RT-PCR positive samples with Ct-values of 34 or less are included in the evaluation (N=103)”. It is not clear if all the 103 samples have a Ct of 35 or less.

At line 118 the authors stated that a total of 999 samples were tested between December 15th 2021 and January 5th 2022 but this number is not in accordance with the total of samples indicated in table 4 (951 persons). There is a confusion between number of samples and person numbers.

In table 4 the authors mention that “SARS-CoV-2 PCR-positive samples were selected for inclusion in this analysis based on viral Ct ≤ 30 (N=91)” but also in this case there is a confusion about Ct value used for inclusion respect to what mentioned in the text.

Illumina sequencing is included as a method but no information about Illumina results are mentioned about Omicron lineage. Which types of Omicron lineages were discovered (for example BA.1, BA.2)?

In each of the 4 tables reported in the study the authors stated that there are samples that resulted negative in RT-PCR but positive for the COVISTIX rapid antigen test. No discussion about these results are reported in the discussion paragraph. Please add some sentences in the discussion session about this.

Lines 232-233: the authors stated that “We classified samples as positive for SARS-CoV-2 when primer-probe sets were detected with a Ct-value of less than 40. If only one of these genes was detected, we labeled the sample as inconclusive”. No indications, except for the Ct value, were included about the criteria to consider the positivity of a sample.

Reviewer 2 Report

This article describes a comparison between the COVISTIX antigen Rapid test for SARS-CoV-2 detection and the TaqPath COVID-19  RT-PCR kit. Some changes to the manuscript are suggested.

  1. Line 132-133: “91 individuals were detected among 951 persons”. In table 4, 15 samples tested COVI-STIX + and PCR negative. Were these individuals not considered positive? Since in lines 201-203 is stated: “Individuals were considered as positive for SARS-CoV-2 if either the nasal or the nasopharyngeal swab resulted positive with the COVISTIX rapid antigen test”.
  2. Can you specify the results of the sequencing? Are all 91 SARS-CoV-2 Omicron sequences 100% similar? Is there a difference in sequences between COVI-STIX positive and negative samples?
  3. Why didn’t you include the 15 COVI-STIX + and PCR negative samples (table 4) in the sequencing protocol?
  4. Were the 7 RT-PCR positive COVISTIX negative samples with Ct values <25 (table 2) confirmed discrepant? Do you know the variant of these samples? Is it possible these differences are related to the fact that different nasopharyngeal swabs were taken for each assay?
  5. If initial nasal swab was negative a nasopharyngeal swab was taken. Were both swabs included into the study afterwards? Which percentage of the patients were negative in the initial nasal swabs but positive in the nasopharyngeal swabs?
  6. Are there differences in intensity of the positive results lines of the antigen test between samples positive in both assays and samples which were RT-PCR negative and antigen test positive?
  7. If only one gene was detected in the RT-PCR, samples were labeled as inconclusive(line 234). Were the results of these samples included in the study?
  8. In line 129 is stated there were no significant differences in the sensitivity of the antigen test associated with Ct-values. However, the overall sensitivity of cohort 2 (December 2021-january 5 2021) is 72% (line 121) and when only Ct-values <34 are included 92 or 93%. Can you explain?
  9. I assume 40 samples from cohort 2 had Ct values > Ct 30 and were not sequenced? Since 131 samples were positive (line 118)?
  10. What is de added value of the separate sensitivity for the 91 RT-PCR positive confirmed omicron samples? If the 40 extra positive samples were not typed these samples could still contain the omicron variant.

Reviewer 3 Report

The article describes the performance of antigen test which is relevant to controlling the COVID at the community level settings as well as in the health care settings.

I would suggest the authors include a detailed flowchart of the algorithm used for the antigen/PCR test. It would be useful to implement the antigen test in different settings.

I would suggest including more references, especially in the ‘materials and methods’ section.
